# Surgical Treatment of Intrahepatic Cholangiocarcinoma: Current and Emerging Principles

**DOI:** 10.3390/jcm10010104

**Published:** 2020-12-30

**Authors:** Eliza W. Beal, Jordan M. Cloyd, Timothy M. Pawlik

**Affiliations:** Department of Surgery, Division of Surgical Oncology, The Ohio State University Wexner Medical Center and James Cancer Hospital, Columbus, OH 43201, USA; Eliza.Beal@osumc.edu (E.W.B.); Jordan.Cloyd@osumc.edu (J.M.C.)

**Keywords:** intrahepatic cholangiocarcinoma, biliary tract cancers, surgery, liver transplantation

## Abstract

Intrahepatic cholangiocarcinoma (ICC) is a rare, aggressive cancer of the biliary tract. It often presents with locally advanced or metastatic disease, but for patients with early-stage disease, surgical resection with negative margins and portahepatis lymphadenectomy is the standard of care. Recent advancements in ICC include refinement of staging, improvement in liver-directed therapies, clarification of the role of adjuvant therapy based on new randomized controlled trials, and advances in minimally invasive liver surgery. In addition, improvements in neoadjuvant strategies and surgical techniques have enabled expanded surgical indications and reduced surgical morbidity and mortality. However, recurrence rates remain high and more effective systemic therapies are still necessary to improve recurrence-free and overall survival. In this review, we focus on current and emerging surgical principals for the management of ICC including preoperative evaluation, current indications for surgery, strategies for future liver remnant augmentation, technical principles, and the role of neoadjuvant and adjuvant therapies.

## 1. Introduction

Intrahepatic cholangiocarcinoma (ICC) is a rare aggressive cancer of the biliary tract. It is the second most common primary hepatic malignancy and is occurring with increasing incidence in the United States [1,2]. The last decade has seen several advances in the diagnosis, staging, and management of ICC. The 7th edition of the American Joint Committee on Cancer (AJCC) Staging Manual introduced a distinct staging system for ICC, and the 8th edition provided further refinement of this staging system, leading to more accurate stratification of prognosis [3]. An improved understanding of the genetic underpinnings of ICC has led to identification of new molecular biomarkers and increased opportunities for targeted therapies [4,5]. Refinement of liver-directed therapies have expanded treatment options for patients with locally advanced disease and improved local control [6,7,8]. New randomized controlled trials have clarified the role of adjuvant therapy for biliary tract cancers [9,10,11]. Despite improvements in prognostication and targeted treatment options, overall survival (OS) among patients with ICC remains low with 5-year overall survival of less than 10% [12,13,14]. This is partly related to the fact that the majority of patients with ICC present with either metastatic, or locally advanced, unresectable disease, and effective systemic therapy options are still lacking.

For patients with localized, potentially resectable cancers, surgery remains the mainstay of treatment. Surgical principles include achieving a margin negative resection and performing an oncologic-directed regional lymphadenectomy [15,16,17]. In recent years, efforts to simultaneously expand the number of patients eligible for curative-intent resection and reduce the morbidity of surgery have led to increased use of vascular resection techniques and minimally invasive approaches, respectively [18,19,20,21,22]. Despite the importance of surgical resection to overall prognosis, recurrence is common, occurring in up to 75% of patients after hepatectomy at 5 years, highlighting the importance of developing multimodality therapeutic approaches in order to improve the long-term outcomes for this aggressive cancer [23]. In this review, we focus on the surgical aspects of managing ICC including current and emerging principles.

## 2. Preoperative Evaluation

The National Comprehensive Cancer Network (NCCN) provides guideline recommendations for the workup, diagnosis, and treatment of patients with ICC [24]. Prior to considering intervention, a patient with a liver mass concerning for malignancy should undergo multiphase imaging of the abdomen with computed tomography (CT) or magnetic resonance imaging (MRI) with intravenous (IV) contrast and chest CT with or without IV contrast. Laboratory studies should be obtained, including tumor markers (alpha-fetoprotein (AFP), carcinoembryonic antigen (CEA), and cancer antigen 19-9 (Ca 19-9)). For individuals without evidence of extrahepatic metastatic disease, the patient should be referred for surgical consultation, and care decisions should be made by an experienced, multidisciplinary team. In patients for whom surgical resection is planned, a biopsy may not be necessary, but pathologic diagnosis should be obtained before beginning systemic chemotherapy or radiation [25]. Diagnosis of ICC based on needle/core biopsy can be difficult to distinguish from other adenocarcinomas. In general, histology typically reveals adenocarcinoma with associated biliary dysplasia. Immunohistochemical staining typically is notable for negative staining for lung (TTF1), colon (CDX2), and pancreas (DPC4) markers with positive staining for biliary epithelium (AE1/AE3; CK7+ and CK 20-) markers. As primary ICC can be challenging to document on pathology, a work-up for a primary adenocarcinoma should be considered. Diagnostic tests may include esophagogastric duodenoscopy, colonoscopy, and mammography as appropriate to rule out an unknown primary.

## 3. Diagnostic Imaging of ICC

A variety of imaging techniques are used for the diagnosis and evaluation of ICC. There are three described morphologic sub-types of ICC including mass-forming, periductal-infiltrating, and intraductal-growth subtypes, with the mass-forming type being the most common type of ICC. Morphologic subtype impacts diagnostic sensitivity and specificity of diagnostic imaging modalities [26].

Ultrasound is often the first imaging modality used to evaluate patients with obstructive jaundice or abdominal pain and can rule out choledocholithiasis and identify cholangiocarcinoma. Contrast-enhanced ultrasound can help to distinguish ICC from hepatocellular carcinoma (HCC) with 64.1% sensitivity, 97.4% specificity, and 73.6% accuracy; peripheral rim-like enhancement and quick contrast washout has high efficiency to distinguish ICC from HCC [27]. Mass-forming ICC appears as homogeneous masses with intermediate to increased echogenicity and a peripheral hypoechoic halo. On contrast enhanced ultrasound, mass-forming ICC may mimic HCC with early enhancement and washout [26].

Multiphase contrast-enhanced CT with arterial, portal venous, and delayed phases can be used to identify and characterize liver masses and demonstrate prognostic features including vascular encasement, nodal involvement, and metastatic disease. On triple-phase CT, in the arterial and portal venous phases, ICC remains hypoattenuating with or without rim enhancement relative to liver parenchyma, with enhancement in the delayed phase and gradual centripetal enhancement on dynamic studies [28]. Furthermore, the portal venous phase accentuates the presence of fibrous stroma, a distinguishing feature of ICC [26]. Degree of enhancement on delayed phases helps to distinguish mass-forming ICC from HCC and has prognostic value [29]. Additionally, CT techniques can be used to calculate liver volume and assist in individualizing plans for resection. Shortcomings of CT include exposure to ionizing radiation and more limited ability to detect tumor tracking along bile ducts and reduced sensitivity to detect lymph node involvement compared with MRI [30].

MRI evaluation of ICC should include magnestic resonance cholangiopancreatography (MRCP), conventional T1- and T2-weighted images, diffusion weighted images, and multiphase contrast-enhanced sequences in arterial, portal venous and delayed phases [26]. On MRI mass-forming ICC demonstrates intensity on T2-weighted imaging and low signal intensity on T1-weighted imaging. On contrast-enhanced MRI, ICC exhibit peripheral rim enhancement with centripetal or progressive enhancement [28]. Several features can be used to distinguish ICC from HCC on MRI with a lobulated shape, rim enhancement in the arterial phase, and a target appearance with a peripheral hyperenhancing rim on diffusion-weighted imaging favoring ICC, while intralesional fat, diffuse hyperintensity on T1-weighted images, nodule-in-nodule appearance, and capsular appearance during portal venous or transitional phase favor HCC [28]. Gadoxetic acid-enhanced MRI can lead to more effective discrimination between ICC and HCC [31].

Moreover, 18F-Fluoro-2-deoxy-D-glucose (FDG) positron emission tomography (PET) coupled with CT can also be used to identify and evaluate liver cholangiocarcinoma, with higher sensitivity and specificity for ICC (>90%) compared with extrahepatic cholangiocarcinoma. All morphologic types of ICC are FDG-avid on PET-CT, which may improve nodal staging over MRI and preoperative standardized uptake value may be an independent risk factor for recurrence after resection [32,33].

## 4. Indications for Surgery

Assessment of surgical candidacy for patients with ICC requires a comprehensive evaluation across several domains: physiologic, anatomic, and biologic. Physiologic resectability refers to the patient’s performance status, comorbidities, and overall ability to tolerate a major operation with acceptable risk of morbidity. Anatomic resectability of ICC is defined as the ability to completely remove the diseased portion of the liver, while leaving behind an adequate future liver remnant (FLR) typically defined as at least two contiguous biliary segments with intact hepatic arterial, portal venous, hepatic venous, and biliary drainage of sufficient volume based on prespecified thresholds. When FLR is inadequate, several augmentation strategies are available. Multifocal liver disease is considered to be representative of metastatic disease and is generally considered a relative contraindication to surgery as the primary therapeutic option; in selected cases, resection of limited multifocal disease can be considered, usually after administration of preoperative chemotherapy [24]. Extrahepatic disease, bilobar or multicentric tumor and lymph node metastases beyond the primary area are other contraindications to resection [30]. In particular, it is important to assess the likelihood of benefit from surgery based on underlying tumor biology. Important considerations include the presence of extrahepatic disease; radiologic features such as vascular invasion, multifocality, obvious lymph node involvement, or subtle signs of peritoneal involvement; histopathologic features; degree of Ca 19-9 elevation; response to prior therapies; and genetic and molecular features of the tumor. A large international study of patients who underwent resection of ICC noted that large tumor size, higher number of tumors, microvascular invasion, N1 or NX disease, suspicious/metastatic lymph nodes on preoperative imaging and R1 resection were associated with a higher likelihood of very early recurrence (≤6 months), suggesting some patients may have benefited from neoadjuvant chemotherapy [34].

## 5. Surgical Principles

### 5.1. Diagnostic Laparoscopy

Given the prevalence of occult peritoneal and hepatic metastases, some investigators have proposed the use of diagnostic laparoscopy prior to surgical resection for ICC. Limited studies have examined the role of staging laparoscopy in patients with ICC. Single-center prospective studies noted that 20–36% of ICC patients had metastatic or unresectable disease on laparoscopy [35,36]. In a single-center retrospective study, diagnostic laparoscopy was performed in 22 of 53 patients, with 6 patients demonstrating unresectable disease on laparoscopy (4 peritoneal disease and 2 intrahepatic metastasis). Another 5 patients had unresectable disease that was not demonstrated on laparoscopy; all patients had disease within the celiac lymph nodes that would not have been discovered on laparoscopy [37]. Other studies have reported relatively low yield and have noted that laparoscopy was not cost effective. Therefore, the role of diagnostic laparoscopy in ICC remains limited and poorly defined. Some clinicians perform laparoscopy selectively for high-risk patients (e.g., elevated Ca 19-9, indeterminate radiographic findings) as recommended by an expert consensus statement from the American Hepato-Pancreato-Biliary Association [25]. Given that an increasing number of liver resections are being performed via a minimally invasive approach, the role of “stand alone” diagnostic laparoscopy prior to laparotomy is becoming less a topic of debate.

### 5.2. Resection

A primary aim of any curative-intent surgery for ICC is to achieve a microscopically (R0) negative margin. Incomplete resection (R1/R2) is associated with higher risk of recurrence and worse OS [15]. An international multi-institutional study to identify factors associated with adverse prognosis after resection of ICC noted that positive margin status (hazard ratio [HR], 2.20; *p* < 0.001) was one of the strongest factors associated with worse OS, in addition to multiple lesions (HR, 1.80; *p* = 0.001) and vascular invasion (HR, 1.59; *p* = 0.015) [35]. Furthermore, compared with patients who underwent a resection with margins of ≥ 1 cm, patients who had narrow surgical margins (5–9 mm: HR 1.21; 1–4 mm: HR 1.32) or positive margins (HR 1.87; *p* = 0.002) had worse recurrence-free survival (RFS). A similar trend was demonstrated for OS (5–9 mm: HR 1.21; 1–4 mm HR 1.95; positive margin HR 2.16, *p*-trend < 0.001) [38]. Treatment at an academic center has also been associated with a lower incidence of a positive surgical margin, as well as decreased 90-day mortality, and improved OS among patients who underwent ICC resection [1,39].

In recent years, there has been increasing focus on parenchymal-sparing approaches to liver surgery. While the role of nonanatomic resections for colorectal liver metastases and HCC has been well-studied, the role of nonanatomic resection for ICC remains uncertain [40,41,42]. For example, in a single-center study in China, Si et al. examined the outcomes among 915 patients who underwent liver resections for ICC and determined that anatomic resection was associated with improved survival outcomes compared with nonanatomic resection in ICC patients with stage IB or II tumors without vascular invasion [43]. In turn, surgical resection does not necessarily need to include an anatomic resection; rather a parenchymal-sparing approach appears acceptable as long as an R0 margin can be achieved with a surgical margin preferably 5–10 mm in width [44].

In order to achieve a margin-negative resection, complex vascular resection may be needed. Vascular resection for ICC is feasible and can be performed safely in select patients at high volume, experienced centers (Figure 1 and Table 1, Selected References) [20,21,45]. In a multi-institutional study of patients who underwent resection for ICC, patients who had a major vascular resection had comparable perioperative (any complication and major complication) and oncologic (RFS and OS) outcomes compared with patients who did not require a vascular resection. While concurrent major vascular resection should be considered in appropriately selected patients with ICC undergoing hepatectomy, these operations should be performed at high volume centers [20]. Similarly, occasionally ICC will involve the proximal or contralateral hepatic ducts and biliary reconstruction will be necessary [44]. In general, complex hepatic resections are indicated to facilitate complete surgical resection with achievement of an R0 margin.

### 5.3. Lymphadenectomy

Lymph node status is one of the most important prognostic factors for localized ICC. Routine regional lymphadenectomy at the time of hepatic resection is, therefore, a critically important component of staging to help determine prognosis, guide adjuvant therapy decisions, and potentially improve locoregional control (Figure 2) [46,47]. The National Comprehensive Cancer Network (NCCN) recommends removal of at least six lymph nodes [24]. Nevertheless, the routine use of lymphadenectomy has remained relatively low and has not increased over time. To this point, Zhang et al. examined utilization of lymphadenectomy in the surgical management of ICC across the United States using the SEER database from 2000 to 2013 and noted that use of lymphadenectomy remained relatively low—with 41% of patients having 1–5 lymph nodes examined and only 11.4% having ≥6 lymph nodes examined; the utilization of lymphadenectomy and the number of nodes harvested did not change over time. While some clinicians have proposed a selective approach to lymphadenectomy for high risk patients, selective utilization of lymph node dissection may be problematic as T-stage has not been a reliable predictor of nodal status—with almost a quarter of patients with early-stage disease having lymph node metastasis [16].

A standard porta hepatis lymphadenectomy should include all lymph nodes along the common hepatic artery (station 8) and within the hepatoduodenal ligament (station 12) [48]. There are additional at-risk lymph node basins based on tumor locality and standard lymphatic drainage patterns. For example, patients with ICC in the left and right hemiliver may benefit from lymphadenectomy along the lesser curvature of the stomach and retropancreatic region, respectively, in addition to the nodes in the hepatoduodenal ligament [49]. In a study of 15 high-volume centers worldwide, Zhang et al. demonstrated that examination of ≥6 lymph nodes had the greatest discriminatory power for OS relative to patients with 1–2 lymph nodes examined. Among patients who underwent an adequate lymphadenectomy with ≥6 lymph nodes examined, lymph node metastasis beyond the hepatoduodenal ligament was associated with worse OS versus lymph node metastasis within the hepatoduodenal ligament only [17].

### 5.4. Minimally Invasive Surgery

While the application of minimally invasive techniques to liver surgery has been relatively slower versus other specialties, the use of minimally invasive liver resection (MILR) has expanded significantly in recent years [19,50]. In a single-center report of 1062 patients undergoing laparoscopic liver resection (LLR) between 2001 and 2017, mean operating time, transfusions, and postoperative complications decreased. Of note, the rate of unplanned open conversion was low (2.5%) and did not change over time; 30- and 90-day mortality was also low at 0.2% and 0.4%, respectively. In addition performance and perioperative morbidity associated with LLR improved with increased experience [51]. A systematic review comparing LLR to open liver resection (OLR) for benign and malignant etiologies demonstrated that there was no increased mortality and fewer complications, transfusions, blood loss, and hospital stay observed in LLR versus OLR; overall morbidity and mortality were 0.3% and 10.5%, respectively [52,53]. The OSLO-COMET trial randomized 280 patients with resectable liver metastasis from colorectal cancer to LLR or OLR using a parenchymal sparing approach and demonstrated that patients who underwent LLR had lower postoperative complications (LLR 19%, OLR 31%, *p* = 0.021) and shorter postoperative hospital stay (53 versus 96 h, *p* < 0.001). There were no differences in blood loss, operation time, resection margins, 90-day mortality, or 4-month cost [54].

Data on MILR relative to ICC have been more limited and of lower quality (Table 2). Single-center studies have demonstrated no differences in perioperative morbidity, mortality, or long-term outcomes including recurrence-free survival (RFS), disease-free survival (DFS), and OS [55,56,57,58,59,60]. In a systematic review, Shiraiwa et al. reported on surgical and oncologic outcomes of LLR for ICC. LLR was associated with lower blood loss and less need for the Pringle maneuver; however, OLR was used more often for major hepatic resection, and was associated with a higher utilization of lymphadenectomy and a higher number of harvested lymph nodes. Unfortunately, most data on the use of MILR for ICC suffer from high heterogeneity and selection bias [61]. Notwithstanding these limitations, another systematic review and meta-analysis, which included 6 retrospective studies including 384 patients who had undergone LLR and 2147 patients who underwent OLR for ICC, noted that patients who underwent LLR more commonly had an R0 resection [62]. However, similar to previous reports, LLR was less often utilized for major hepatectomy and lymph node dissection was lower among patients who had a minimally invasive approach. As such, while LLR had comparable safety, feasibility, and oncologic efficacy to that of OLR for ICC, the data were limited and further studies are needed to evaluate surgical approach among patients in need of major hepatic resection [60]. In particular, there is a significant learning curve related to MILR, especially for major hepatectomy, and thus, these operations should be performed at high-volume institutions by experienced surgical teams [63,64].

Other technologies have been sought to improve the ability to achieve a margin negative resection while preserving the FLR, including the use of real-time navigation with indocyanine green fluorescence imaging systems [65]. Additionally, robotic surgical techniques have been increasingly applied to liver resection. Meta-analyses comparing LLR to robotic liver resection demonstrated no differences in conversion to open hepatectomy, margin-positive resection, blood loss, transfusion requirement, operative time, length of stay, overall complications, severe complications, or overall mortality [66,67].

### 5.5. Future Liver Remnant Augmentation Strategies

A subset of patients being considered for resection will have an inadequate FLR, which increases the risk of posthepatectomy liver failure and subsequent mortality. FLR can be assessed volumetrically using cross-sectional imaging or functionally with the indocyanine green clearance test or other novel functional imaging tests [68]. Previous studies have established the minimal FLR volume, measured as a percentage of calculated standardized total liver volume to minimize the risk of posthepatectomy liver failure: 20% with normal liver function, 30% with hepatic steatosis or chemotherapy associated liver injury, and 40–50% with severe fibrosis or cirrhosis [69]. When the FLR is inadequate, several strategies for FLR augmentation are available including portal vein embolization (PVE), liver venous deprivation, and associating liver partition and portal vein ligation (ALPPS).

PVE is an effective strategy to induce contralateral liver hypertrophy prior to major liver resection. The underlying principle of PVE includes interrupting portal venous blood flow to liver segments planned for resection. PVE leads to atrophy of the segments without portal flow, with accompanying reactive hypertrophy of the FLR. With growing radiologic capabilities, percutaneous transhepatic PVE has become the standard technique for portal vein occlusion. A recent systematic review included 44 articles (*n* = 1791 patients) on outcomes of PVE. The overall technical success rate was very high at 99.3%, and the mean hypertrophy rate of the FLR was 37.9 ± 0.1% [70].

Liver venous deprivation is another FLR augmentation strategy that combines PVE and hepatic vein embolization in the same procedure [71]. Single-center studies have demonstrated that liver venous deprivation results in reasonable FLR hypertrophy, facilitates R0 resection and results in minimal morbidity/mortality [72]. Extended liver venous deprivation with embolization of the right portal vein, right hepatic vein, and middle hepatic vein has also been described [73]. Comparing liver venous deprivation to PVE, single-center studies have noted that ipsilateral venous deprivation before major hepatectomy induces similar or greater/faster FLR hypertrophy than after PVE alone with similar morbidity and mortality [71,74,75]. Both PVE and liver venous deprivation can be considered when assessing patients with ICC who are in need of an extended liver resection.

Associating liver partition and portal vein ligation (ALPPS) is an alternative two-stage approach to induce rapid FLR hypertrophy. In the standard ALPPS procedure, the first operation involves intraoperative ligation of the right portal vein as well as transection of the liver parenchyma along the planned resection line leaving the hepatic arterial and venous branches intact. Completion of the resection is performed at a second operation once sufficient FLR hypertrophy is confirmed with volumetry, typically 1–2 weeks later [76]. An international multi-institutional study of ALPPS included 102 patients with ICC; despite high 90-day mortality, ALPPS was associated with high rates of achieving R0 resection (N = 87, 85%) in locally advanced ICC with improved OS compared with palliative chemotherapy alone (median OS: 26.4 months versus 14 months; 1-, 2-, and 3-year survival: 82.4%, 70.5%, and 39.6% versus 51.2%, 21.4%, and 11.3%, respectively, *p* < 0.01 [77]. Nevertheless, the use of ALPPS remains controversial given its relatively high morbidity and mortality and should be limited to high-volume experienced institutions [76]. Novel modifications to this procedure (e.g., mini-ALPPS) may reduce the morbidity while expanding the indications for surgery [78].

### 5.6. Transplantation for ICC

Intrahepatic cholangiocarcinoma is currently considered a contraindication for liver transplant (LT) in most centers secondary to poor outcomes in the early LT for ICC experience [79]. Several studies included patients with both hilar and intrahepatic cholangiocarcinoma, as well as patients with and without cirrhosis. Attempts have been made more recently to analyze more homogeneous cohorts. A matched cohort multicenter study in Spain evaluated outcomes of patients with hepatocellular-cholangiocarcinoma (HCC-CC) or ICC on pathologic exam after LT compared with patients who had HCC. ICC patients had 5-year OS of 50%; patients with larger tumors (>2 cm) were at higher risk of worse outcomes versus patients with comparable HCCs, while patients with small, single ICC had similar results to patients with HCC [80]. A separate study from this same research group demonstrated that patients with “very early” ICC, including tumors ≤ 2 cm, who underwent LT had a 5-year OS of 73%. In contrast, patients with tumors > 2 cm and/or multifocal disease had 5-year OS of 40%. This study was small, however, as it included only 29 patients [81]. An additional study compared “very early” ICC patients (N = 15, ≤ 2 cm) to patients with “advanced” disease (N = 33, >2 cm, multifocal) and noted a 5-year actuarial survival of 65% and 45% and 5-year recurrence of 18% and 61%, respectively [81]. Of note, the aforementioned studies included patients who were thought to have HCC and instead had ICC on explant pathology or were transplanted for advanced cirrhosis and were incidentally diagnosed with ICC on pathology. The authors noted that these data need to be prospectively reproduced and that LT for ICC should be reserved for patients with cirrhosis and portal hypertension for whom liver resection is not an option. Very small prospective protocols have been pursued including neoadjuvant chemotherapy followed by LT for ICC patients; these studies concluded that selected patients with stable, locally advanced ICC on neoadjuvant therapy may benefit from LT [82]. Based on these data, a single-arm clinical trial (NCT02878473) to examine the role of LT for patients with very early (single lesion, ≤2 cm) ICC is ongoing [83].

## 6. Perioperative Therapies

### 6.1. Adjuvant Therapy

Patients with ICC who undergo curative-intent resection still have a high incidence of recurrence; therefore, adjuvant therapy should be considered [24]. In particular, patients with a high tumor burden are at particularly high risk of recurrence and may be in need of postresection therapy [84]. Several recent trials have helped to clarify the role of adjuvant therapy for patients with ICC. The PRODIGE 12-ACCORD 18 trial was a multicenter, open-label, phase III trial that randomized patients to adjuvant chemotherapy with gemcitabine/oxaliplatin (GEMOX) versus observation following R0/1 resection for biliary tract cancers (BTC) including ICC (43%). The trial demonstrated no difference in health-related quality-of-life, relapse-free survival or OS [11]. The investigators concluded that, while the regimen was well tolerated, there was no benefit to adjuvant GEMOX among patients with resected BTCs [11]. The BILCAP trial was a randomized, controlled, multicenter, phase III study in which patients with R0/R1 resected cholangiocarcinoma or muscle-invasive gallbladder cancers were randomized to oral capecitabine or observation. Of note, only 84 of the 447 (19%) of the patients included in the trial had intrahepatic cholangiocarcinoma. Although the study did not meet its primary end point of improving OS in the intention-to-treat analysis, capecitabine was associated with improved OS in the prespecified sensitivity and perprotocol analyses (53 months versus 36 months, *p* = 0.028) [9]. Based on the results of BILCAP trial, adjuvant therapy after resection of ICC should be considered, but further studies are still needed to clarify its’ role. The ACTICCA-1 trial is a randomized, multidisciplinary, multinational phase III trial in which patients with biliary tract cancers (BTCs) were randomized to gemcitabine and cisplatin versus observation; this study is ongoing [10]. The JCOG1202, ASCOT trial is an open-label, multicenter, randomized phase III trial randomizing patients with resected BTCs to S-1 therapy versus observation and is also ongoing [85].

### 6.2. Neoadjuvant Therapy

While debated, there is a sound rationale for the delivery of neoadjuvant chemotherapy in ICC. Preoperative chemotherapy may help downstage locally advanced tumors, improve margin negative resection, increase receipt of systemic therapy, prioritize the early systemic treatment of potential micrometastatic disease, as well as enhance patient selection for major surgery and facilitate in vivo test of chemotherapy effectiveness [86]. While no prospective randomized controlled trials of neoadjuvant chemotherapy have been conducted for patients with BTCs, increasing retrospective data highlight its utility to treat a subset of patients with locally advanced disease [87]. A single-center study included 74 patients with locally advanced ICC, 53% (N = 39) of whom underwent secondary resection after a median of six cycles of chemotherapy using various chemotherapeutic regimens and locoregional approaches. These patients were compared to a group of patients who underwent upfront resection. The secondary resection group patients were younger, had more advanced disease and more commonly had lymphadenopathy and vascular invasion. The authors found no difference in postoperative morbidity, mortality or median survival (24.1 versus 25.7 months, *p* = 0.391) between groups. The authors concluded that neoadjuvant chemotherapy may be an effective downstaging option for patients with locally advanced ICC [88]. Neoadjuvant chemotherapy regimens usually include gemcitabine and cisplatin, based on the ABC-02 trial, which demonstrated the improved efficacy of combination therapy over gemcitabine alone [89]. Locoregional approaches have been successfully used both for palliative treatment and for downstaging of ICC prior to resection including transarterial chemoembolization, combination of Yttrium-90 radioembolization and systemic chemotherapy, and selective internal radiation therapy with or without chemotherapy [7,90,91,92]. Downstaging locally advanced ICC prior to LT has also been reported [93].

## 7. Management of Recurrent ICC

Treatment for recurrent ICC is varied and a few studies have examined outcomes following re-resection of recurrent ICC. In a multi-institutional study that included 563 patients with ICC who underwent surgical resection, 400 (71.0%) individuals developed intrahepatic only (59.8%), extrahepatic only (14.5%), or intra- and extrahepatic (25.7%) recurrence with an overall median DFS of 11.2 months. Among individuals who experienced a recurrence, 210 (52.5%) patients received best supportive care (BSC), while 190 (47.5%) underwent treatment with liver-directed therapy plus systemic chemotherapy (N = 144, 75.8%) or systemic chemotherapy only (N = 46, 24.2%). Patients who underwent repeat liver-directed therapy had hepatic resection ± ablation (28.5%), ablation alone (18.7%), or intra-arterial therapy (IAT, 52.8%). Among patients who underwent resection of recurrent ICC ± ablation median survival was 26.1 months, which was comparable to median survival among patients who underwent ablation-alone (25.5 months), yet lower than patients who had IAT (9.6 months). Among patients who underwent repeat resection ± ablation, 53.6% of patients had a second recurrence with median time to recurrence of 11.6 months. The authors concluded that recurrence was common and that repeat liver resection after recurrence was feasible only in a small subset of patients with modest survival benefit [94]. In a different multi-institutional study of 356 patients with ICC, 214 (60%) had a recurrence after surgical resection; 37 (17%) of these patients underwent subsequent surgical treatment for the recurrence with an associated 5-year OS of 44% [95]. Single-center studies comparing patients who underwent resection for recurrence ICC versus best supportive care or other liver-directed therapies/chemotherapy have demonstrated improved OS for resection; however, these studies need to be interpreted carefully due to significant selection bias and confounding due to the retrospective study design [96,97,98,99,100].

## 8. Locoregional Therapy for Unresectable ICC

Although surgical resection is the cornerstone of treatment for ICC, a variety of locoregional treatments are used in the setting of unresectable ICC, or in patients who are not candidates for surgery. The more commonly used locoregional approaches include radiofrequency ablation (RFA), transarterial chemoembolization (TACE), and transarterial radioembolization (TARE) with Yttrium-90. The goals of locoregional therapy include controlling local tumor growth, relieving symptoms and improving or preserving quality of life.

RFA can be delivered percutaneously with CT-guidance or intraoperatively with ultrasound-guidance. During image-guided ablation thermal energy is delivered to the tumor tissue through needle electrodes to reach intratumoral temperatures of 60–100 °C to induce tumor necrosis. RFA can be used as primary treatment for unresectable ICC or in the setting of recurrent ICC [101,102,103]. In a systematic review and meta-analysis of clinical efficacy and safety of RFA in the treatment of ICC, seven observational studies including 84 patients were reviewed. The authors demonstrated 1-year, 3-year, and 5-year survival of 82%, 47%, and 24%, respectively, and concluded that RFA is a locoregional treatment option that prolongs survival in patients with ICC [102]. A recent single-center study examining local tumor progression-free survival for patients undergoing RFA included 29 patients with 117 nodules. The authors demonstrated that on univariate and multivariate analysis, tumor size ≥ 2 cm was associated with reduced local tumor progression-free survival and suggest that 2 cm may represent a useful threshold value [104].

TACE is the most commonly used intra-arterial therapy for ICC patients and is performed using an emulsion of chemotherapy and oil-based contrast agents such as Ethiodol or lipiodol followed by an embolization agent injected into the branch of the hepatic artery supplying the tumor. Commonly used chemotherapeutic agents include doxorubicin, cisplatin, and mitomycin-C. It has been demonstrated that for patients with ICC, TACE is able to improve survival when compared to best supportive care. In a systematic review and meta-analysis of patients with unresectable ICC including 542 patients, overall survival of 15.7 months from diagnosis was demonstrated, and 76.8% of patients had response or stable disease on postprocedure imaging with 30-day mortality of 0.7%. Moreover, 19% of patients experienced severe toxicities, and almost all reported post-embolization syndrome including nausea, abdominal pain, fever, and a transient increase in liver enzymes [105]. Future directions for TACE in ICC include the addition of targeted therapies, such as tyrosine kinase inhibitors [106].

TARE with Yttrium-90 aims to selectively release high-dose radiation specifically to liver tumors, while limiting radiation exposure to the normal live parenchyma and surrounding structures. In a single-center, prospective, cohort study, 48 Y-90 treatments were administered to 24 patients with ICC. Further, 22 patients had imaging follow-up—6 (27%) with partial response, 15 (68%) with stable disease, and 1 (5%) with progressive disease. Moreover, 17 (77%) patients demonstrated >50% tumor necrosis, while 2 (9%) had 100% tumor necrosis based on imaging criteria. Median overall survival was 14.9 months among all patients and was higher at 31.8 months among patients without portal vein thrombosis. The majority of patients experienced transient abdominal pain (N = 18, 75%), and fatigue (N = 10, 42%) was also common. Additionally, 1 (4%) patient developed grade 3 bilirubin toxicity, and 1 (4%) developed a treatment-related gastroduodenal ulcer [107]. In a systematic review and meta-analysis including 12 studies and 298 patients, overall weighted median survival was 15.5 months. A weighted mean partial response was seen in 28% and stable disease in 54% of patients at 3 months based on imaging criteria. The most commonly reported morbidities were fatigue (33%), abdominal pain (28%), and nausea (25%). The authors concluded that TARE with Yttrium-90 should be included in the list of potential treatment options for ICC, but that further studies would be needed to determine optimal treatment modalities for unresectable ICC [108].

## 9. Conclusions

ICC is an aggressive malignancy that often presents with locally advanced or metastatic disease. Preoperative work-up should include cross-sectional imaging with CT and MRI to assist with surgical planning. For patients with localized ICC, the primary treatment strategy should involve a margin negative hepatic resection with a porta hepatis lymphadenectomy. Preoperative chemotherapy should be considered for individuals who present with advanced tumor characteristics such as bilateral multifocal disease or clinically metastatic lymph nodes. Following surgery, irrespective of margin or nodal status, postoperative adjuvant chemotherapy should be considered in light of the recent BILCAP data. ICC is a complex disease that requires a multimodality, multidisciplinary approach. Ongoing advances are needed relative to chemotherapy, targeted therapies, as well as locoregional treatments to increase the number of patients who are surgical candidates and, thereby, ultimately improve the long-term outcomes of patients with ICC.

## Figures and Tables

**Figure 1 jcm-10-00104-f001:**
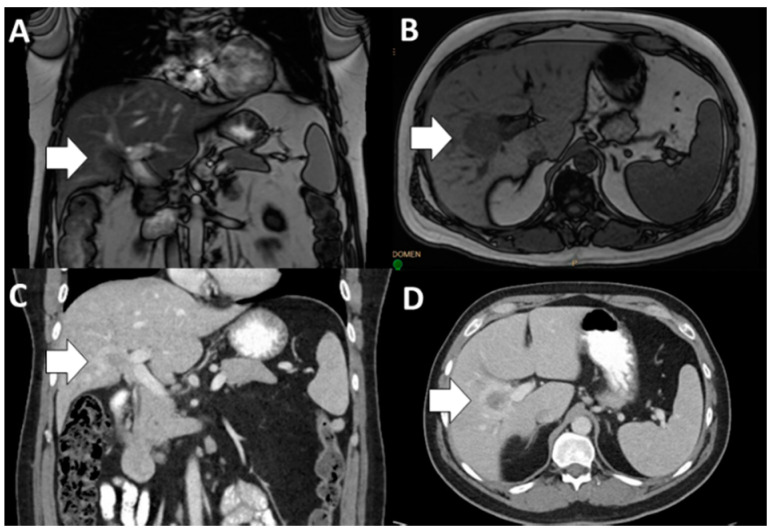
Infiltrative intrahepatic cholangiocarcinoma (ICC) (short, white arrow) with involvement of the right portal vein on magnetic resonance imaging (MRI; (**A**), coronal and (**B**), axial) and computed tomography (CT; (**C**), coronal and (**D**), axial). ICC, intrahepatic cholangiocarcinoma; MRI, magnetic resonance imaging; CT, computed tomography.

**Figure 2 jcm-10-00104-f002:**
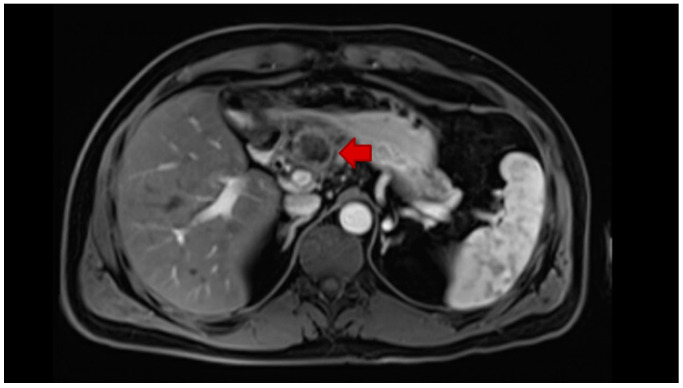
Gastrohepatic lymph node recurrence (short red arrow) in a patient who previously underwent left hepatectomy without porta hepatis lymphadenectomy for ICC.

**Table 1 jcm-10-00104-t001:** Selected references: vascular resection for intrahepatic cholangiocarcinoma.

Author	Study Period	Inclusion	N	Type of VR	Short Term Outcomes	Long-Term Outcomes
Ali et al. [45]	1997–2011	Major hepatectomy, single institution	121; 14 (11.6%) VR	N = 5 PV, N = 5 IVC	Major complications (CV ≥ 3) VR N = 4, 29% versus NVR N = 17, 16%; *p* = 0.263	Median OS VR 32 mo versus NVR 49 mo; *p* = 0.268
Reames et al. [20]	1990–2016	Hepatectomy, 13 institutions; 61.1% major	1087, 128 (11.8%) VR	N = 21, 16.4% IVC; N = 98, 76.6% PV; 9, 7.0%, combined	VR not associated with complications (OR = 0.68, NS), major complications (OR = 0.95, NS), postoperative mortality (OR = 1.05, NS)	Median RFS VR 14.0 mo versus NVR 14.7 mo, HR 0.74, *p* > 0.05; Median OS VR 33.4 mo versus NVR 40.2 mo, HR = 0.71, *p* > 0.05
Conci et al. [21]	1995–2015	Resected ICC, 62.2% major	270, 31 (11.5%) VR	N = 15, 5.6% PVR; N = 16, 5.9% CVR	Postoperative mortality NVR (N = 6, 2.5%) versus VR (N = 3, 9.7%, *p* = 0.072); Major complications NVR (N = 9, 29%) versus VR (N = 39, 16.3%, *p* = 0.082)	PVR, HR 1.57, 95% CI 0.71–3.51, *p* = 0.347; CVR, HR 1.94, 95% CI 0.87–3.85, *p* = 0.238

ICC, intrahepatic cholangiocarcinoma; CV, Clavien-Dindo; VR, vascular resection; NVR, Nonvascular resection; OS, overall survival; PV, portal vein; IVC, inferior vena cava; RFS, recurrence-free survival; PVR, portal vein resection; CVR, IVC resection; NS, not significant; mo, months; HR, hazard ratio.

**Table 2 jcm-10-00104-t002:** Selected references: laparoscopic versus open liver resection for intrahepatic cholangiocarcinoma.

Author	Year	LLR N	OLR N	Short-Term Outcomes	Long-Term Outcomes
Wu et al. [55]	2020	18	25	LLR decreased EBL, LOS; no difference 30-day morbidity or mortality	No difference median, 1-or 3-year RFS/OS
Haber et al. [56]	2020	27	31	LAD LLR 85%/OLR 94%; R0 LLR 89%/OLR 74%; hospital LOS LLR 10/OLR 12 days; overall complications LLR 30%/OLR 58%, no difference in major complications	
Ratti et al. [57]	2016	20	60	EBL LLR 200/OLR 350 mL, despite less use of Pringle; No difference perioperative morbidity or mortality; Functional recovery LLR 3/OLR 4 days	No difference in DFS/OS; Number of harvested nodes comparable
Zhu et al. [58]	2019	20	63	Large (≥5 cm)/multiple (≥2) ICC; After PSM—OR time LLR 225/OLR 190 min; occlusion time LLR 50/OLR 35 min; LLR completed in 18; no difference proportion major liver resection, EBL, transfusion rate, major complications, hospital LOS	No difference in recurrence rate, DFS/OS between LLR and OLR
Uy et al. [59]	2015	11	26	EBL OLR > LLR; no difference in transfusions, OR time, resection margin, LOS	No difference in 3- or 5-year OS/DFS
Lee et al. [60]	2016	14	23	LLR less frequent Pringle maneuver, blood loss; no differences in complication rate, hospital stay, tumor size, lymph node metastasis, number of retrieved lymph nodes	No difference in 3-year OS or RFS

LLR = laparoscopic liver resection; OLR = open liver resection; EBL = estimated blood loss, LOS = length of stay; RFS = recurrence-free survival; OS = overall survival; LAD = lymphadenectomy; DFS = disease-free survival; PSM = propensity score matching; OR, operating room; Min, minutes.

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
