# Peer review of "Surgical Treatment of Intrahepatic Cholangiocarcinoma: Current and Emerging Principles"

_jcm, 2020, doi:10.3390/jcm10010104_

Round 1
Reviewer 1 Report
Dear Editor, thank you so much for inviting me to revise this manuscript about the topic of surgical treatment of intrahepatic cholangiocarcinoma.
In this landscape, the study by Beal and colleagues aimed to discuss surgical approaches for intrahepatic cholangiocarcinoma, thus addressing a current topic.
The manuscript is quite well written and organized.
Figures and tables are comprehensive and clear.
The introduction explains in a clear and coherent manner the background of this study.
We suggest the following modifications:
- Introduction section: although the authors correctly included important papers in this setting, we believe a study should be cited within the introduction (doi: 10.1038/nrclinonc.2017.157), only for a matter of consistency. We think it might be useful to introduce the topic of this interesting study. In addition, we believe the authors should more extensively discuss the recent results of the phase 3 BILCAP trial conducted by Primrose and colleagues, a pivotal trial in this setting. In addition, since randomized controlled trials such as the PRODIGE12-ACCORD18 - assessing adjuvant gemcitabine – oxaliplatin (GEMOX) versus placebo in completely resected biliary tract cancer patients showed no differences in terms of relapse-free survival and overall survival, this issue deserves greater discussion.
- The authors should add more information regarding neoadjuvant approaches. For example, the authors should report some details regarding these two Japanese and French studies: doi: 10.1245/s10434-012-2312-8; doi: 10.1002/bjs.10641. Unfortunately, surgery remains the only curative therapeutic option for iCCA and limited data are available on neoadjuvant treatment, especially in terms of R0 resection rate. In addition, data are mainly limited to small size cohort studies with different treatment modalities - including radiotherapy, chemotherapy, chemoradiation, and local liver-directed therapies - and heterogeneous approaches, with an R0 resection rate ranging from 30 to 80%. We suggest discussing these points more extensively.
- Lastly, although this topic is not exactly part of the paper, the authors should briefly report some details regarding local treatments in iCCA, and in particular radiofrequency ablation, since there is growing attention towards this technique. For example, the authors should include the results of a recent retrospective Italian study on this topic in iCCA, suggesting the association between tumor size larger than 2 cm and poorer progression-free survival (doi: 10.1080/02656736.2020.1763484); thus, 2 cm could represent a useful threshold value for radiofrequency ablation, with clinical outcomes similar to what achieved with surgery.
The main strengths of this paper are that it addresses an interesting and very timely question and provides a clear answer, with some limitations.
Certainly, some changes are required in some parts regarding medical treatment and local therapies.
We suggest the addition of some references and to better clarify some points regarding local approaches, adjuvant and neoadjuvant therapy.
Author Response
Dear Editors and Reviewers,
Thank you for taking the time to review our manuscript, “Surgical Treatment of Intrahepatic Cholangiocarcinoma: Current and Emerging Principles.”. We have made changes that serve to strengthen the manuscript. Please see below for a point-by-point response to reviewer comments.
Thank you,
Timothy M. Pawlik, MD, MPH, PhD, FACS, FRACS (Hon.)
Review 1:
Dear Editor, thank you so much for inviting me to revise this manuscript about the topic of surgical treatment of intrahepatic cholangiocarcinoma. In this landscape, the study by Beal and colleagues aimed to discuss surgical approaches for intrahepatic cholangiocarcinoma, thus addressing a current topic. The manuscript is quite well written and organized. Figures and tables are comprehensive and clear. The introduction explains in a clear and coherent manner the background of this study.
We suggest the following modifications:
Introduction section: although the authors correctly included important papers in this setting, we believe a study should be cited within the introduction (doi: 10.1038/nrclinonc.2017.157), only for a matter of consistency. We think it might be useful to introduce the topic of this interesting study.
We added a citation of, “Cholangiocarcinoma — evolving concepts and therapeutic
strategies (doi: 10.1038/nrclinonc.2017.157),” in the introduction (page 1, line 28).
In addition, we believe the authors should more extensively discuss the recent results of the phase 3 BILCAP trial conducted by Primrose and colleagues, a pivotal trial in this setting. In addition, since randomized controlled trials such as the PRODIGE12-ACCORD18 - assessing adjuvant gemcitabine – oxaliplatin (GEMOX) versus placebo in completely resected biliary tract cancer patients showed no differences in terms of relapse-free survival and overall survival, this issue deserves greater discussion.
We have added additional detail on the BILCAP trial on page 9 lines 341-349. We have also added the sentence, “Given the results of these trials, the role of adjuvant therapy for intrahepatic cholangiocarcinoma remains unclear.”
The authors should add more information regarding neoadjuvant approaches. For example, the authors should report some details regarding these two Japanese and French studies: doi: 10.1245/s10434-012-2312-8; doi: 10.1002/bjs.10641. Unfortunately, surgery remains the only curative therapeutic option for iCCA and limited data are available on neoadjuvant treatment, especially in terms of R0 resection rate. In addition, data are mainly limited to small size cohort studies with different treatment modalities - including radiotherapy, chemotherapy, chemoradiation, and local liver-directed therapies - and heterogeneous approaches, with an R0 resection rate ranging from 30 to 80%. We suggest discussing these points more extensively.
We have added further discussion on neoadjuvant approaches to the section, “Neoadjuvant Therapy,” on page 10. As this was not the focus of this article, we kept the discussion brief.
The study, “Neoadjuvant chemotherapy for initially unresectable intrahepatic cholangiocarcinoma,” by Le Roy et al (doi: 10.1002/bjs.10641) was added to the section on neoadjuvant therapy and some details of the study provided.
The study, “Surgical resection after downstaging chemotherapy for initially unresectable locally advanced biliary tract cancer: a retrospective single-center study,” by Kato et al (doi: 10.1245/s10434-012-2312-8) was added as a citation to the sentence, “While no prospective randomized controlled trials of neoadjuvant chemotherapy have been conducted for patients with BTCs, increasing retrospective data highlight its utility to treat a subset of patients with locally advanced disease,” but the details of the study were not added as it includes only 7 patients with ICC.
Lastly, although this topic is not exactly part of the paper, the authors should briefly report some details regarding local treatments in iCCA, and in particular radiofrequency ablation, since there is growing attention towards this technique. For example, the authors should include the results of a recent retrospective Italian study on this topic in iCCA, suggesting the association between tumor size larger than 2 cm and poorer progression-free survival (doi: 10.1080/02656736.2020.1763484); thus, 2 cm could represent a useful threshold value for radiofrequency ablation, with clinical outcomes similar to what achieved with surgery.
We have added further discussion on locoregional therapies including TACE, TARE with Yittrium-90 and radiofrequency ablation on page 11. We have also included the mentioned study, “Percutaneous radiofrequency ablation in intra-hepatic cholangiocarcinoma: a retrospective single-center experience.”
The main strengths of this paper are that it addresses an interesting and very timely question and provides a clear answer, with some limitations. Certainly, some changes are required in some parts regarding medical treatment and local therapies. We suggest the addition of some references and to better clarify some points regarding local approaches, adjuvant and neoadjuvant therapy.
Thank you, we have incorporated your suggested changes.
Reviewer 2 Report
Thank you for letting me review the manuscript entitled "Surgical Treatment of Intrahepatic Cholangiocarcinoma: Current and Emerging Principles" by Beal et al., in which the authors summarize current developments in the perioperative treatment of this malignant disease.
Needless to say, Pawlik and colleagues are well-known experts on this important matter and the manuscript is well written and organized. However, I find the tables a bit cluttered and the figures dispensable.
Author Response
Dear Editors and Reviewers,
Thank you for taking the time to review our manuscript, “Surgical Treatment of Intrahepatic Cholangiocarcinoma: Current and Emerging Principles.”. We have made changes that serve to strengthen the manuscript. Please see below for a point-by-point response to reviewer comments.
Thank you,
Timothy M. Pawlik, MD, MPH, PhD, FACS, FRACS (Hon.)
Reviewer 2:
Thank you for letting me review the manuscript entitled "Surgical Treatment of Intrahepatic Cholangiocarcinoma: Current and Emerging Principles" by Beal et al., in which the authors summarize current developments in the perioperative treatment of this malignant disease.
Needless to say, Pawlik and colleagues are well-known experts on this important matter and the manuscript is well written and organized. However, I find the tables a bit cluttered and the figures dispensable.
Table 1, on page 5, has been edited to be more concise.
Table 2, on page 7, has been edited to be more concise.
We believe that the figures add some context for the points discussed. We would be happy to remove them if the editors are in agreement.
Reviewer 3 Report
This is a scoping review on intrahepatic cholangiocarcinoma (ICC). The article is well-written and provides good overview about the topic. Below, I present some general and specific remarks to the authors.
- In introduction, authors should provide more objective data. For example, when saying “the overall survival (OS) among patients with ICC remains low” (lines 36-37), readership would benefit to know exactly how the OS is. The same applies when they state that “recurrence is common” (line 46). At the end of the manuscript, they will again state that “ICC is an aggressive malignance” – although the affirmation is correct – I believe the readership would benefit from knowing exactly what the outcomes in ICCA are.
- In section “Preoperative Evaluation”, authors could expand the discussion about the role of preoperative biopsy. The indication of biopsy can be controversial, and it is not necessary for every case. Therefore, authors could describe what is the current role of percutaneous biopsy in ICC.
- Please, check the rates of morbidity and mortality presented on lines 225 and 226.
- When presenting the BILCAP trial, authors are encouraged to mention that only 84 (19% of study population) patients included in study had ICC.
- Abbreviation should be defined at the first time they appear in the manuscript. Also, confider suppressing abbreviations that are not used more than 2 or 3 times throughout the manuscript.
Author Response
Dear Editors and Reviewers,
Thank you for taking the time to review our manuscript, “Surgical Treatment of Intrahepatic Cholangiocarcinoma: Current and Emerging Principles.”. We have made changes that serve to strengthen the manuscript. Please see below for a point-by-point response to reviewer comments.
Thank you,
Timothy M. Pawlik, MD, MPH, PhD, FACS, FRACS (Hon.)
Reviewer 3:
This is a scoping review on intrahepatic cholangiocarcinoma (ICC). The article is well-written and provides good overview about the topic. Below, I present some general and specific remarks to the authors.
In introduction, authors should provide more objective data. For example, when saying “the overall survival (OS) among patients with ICC remains low” (lines 36-37), readership would benefit to know exactly how the OS is. The same applies when they state that “recurrence is common” (line 46). At the end of the manuscript, they will again state that “ICC is an aggressive malignance” – although the affirmation is correct – I believe the readership would benefit from knowing exactly what the outcomes in ICCA are.
We have added, in the introduction (page 1, lines 37-38), that 5-year overall survival for ICC is less than 10%.
We have added, in the introduction (page 2, line 47), that up to 75% of patients recur within 5-years of hepatectomy for ICC.
In section “Preoperative Evaluation”, authors could expand the discussion about the role of preoperative biopsy. The indication of biopsy can be controversial, and it is not necessary for every case. Therefore, authors could describe what is the current role of percutaneous biopsy in ICC.
We have expanded the discussion of the use of biopsy for ICC on page 2 lines 60 – 62 by adding, “In patients for whom surgical resection is planned, a biopsy may not be necessary, but pathologic diagnosis is required before beginning systemic chemotherapy or radiation.”
Please, check the rates of morbidity and mortality presented on lines 225 and 226.
We have confirmed the rates of morbidity and mortality presented on page 6, lines 225-230.
When presenting the BILCAP trial, authors are encouraged to mention that only 84 (19% of study population) patients included in study had ICC.
We have added to our discussion of the BILCAP trial that only 84 (19% of population) patients in the BILCAP trial had ICC on page 9, lines 341-344.
Abbreviation should be defined at the first time they appear in the manuscript. Also, consider suppressing abbreviations that are not used more than 2 or 3 times throughout the manuscript.
We have refined the use of abbreviations throughout the manuscript. For example:
- We removed the abbreviations for ultrasound and contrast-enhanced ultrasound on page 2, lines 76-83.
- We added “hepatocellular carcinoma (HCC)” on page 2, line 78.
- We removed the abbreviation for diffusion weighted images (DWI) on page 3, line 96.
- We removed the abbreviation “SUV” from page 3, line 109.
- We have abbreviated hepatocellular carcinoma as, “HCC,” on page 4, line 166.
- We removed the abbreviation “PHLR” from page 8, line 265 and its’ subsequent uses.
We removed the abbreviations LVD, HVE, eLVD from page 8 and their subsequent uses.
Round 2
Reviewer 1 Report
The authors improved the paper, adding several parts.
We congratulate the authors for their study and we recommend Acceptance in the current form.